# Marine Litter Tracking System: A Case Study with Open-Source Technology and a Citizen Science-Based Approach

**DOI:** 10.3390/s23020935

**Published:** 2023-01-13

**Authors:** Silvia Merlino, Marina Locritani, Antonio Guarnieri, Damiano Delrosso, Marco Bianucci, Marco Paterni

**Affiliations:** 1CNR-ISMAR (Istituto di Scienze Marine-Sede di La Spezia), 19032 La Spezia, Italy; 2Istituto Nazionale di Geofisica e Vulcanologia, Sezione di Roma 2, 00143 Roma, Italy; 3Istituto Nazionale di Geofisica e Vulcanologia, Sezione di Bologna, 40127 Bologna, Italy; 4CNR-IFC (Istituto di Fisiologia Clinica-Pisa), 56124 Pisa, Italy

**Keywords:** IoT technology, citizen science, marine litter, drifters, smart tracking devices, Lagrangian dispersal models

## Abstract

It is well established that most of the plastic pollution found in the oceans is transported via rivers. Unfortunately, the main processes contributing to plastic and debris displacement through riparian systems is still poorly understood. The Marine Litter Drifter project from the Arno River aims at using modern consumer software and hardware technologies to track the movements of real anthropogenic marine debris (AMD) from rivers. The innovative “Marine Litter Trackers” (MLT) were utilized as they are reliable, robust, self-powered and they present almost no maintenance costs. Furthermore, they can be built not only by those trained in the field but also by those with no specific expertise, including high school students, simply by following the instructions. Five dispersion experiments were successfully conducted from April 2021 to December 2021, using different types of trackers in different seasons and weather conditions. The maximum distance tracked was 2845 km for a period of 94 days. The activity at sea was integrated by use of Lagrangian numerical models that also assisted in planning the deployments and the recovery of drifters. The observed tracking data in turn were used for calibration and validation, recursively improving their quality. The dynamics of marine litter (ML) dispersion in the Tyrrhenian Sea is also discussed, along with the potential for open-source approaches including the “citizen science” perspective for both improving big data collection and educating/awareness-raising on AMD issues.

## 1. Introduction

The Mediterranean Sea is a semienclosed basin. It receives waters from several large rivers and it hosts the 20% of the world’s maritime traffic. Its coasts are highly populated, which induces extensive and varied anthropic activities, even if the waste treatment infrastructures are undersized [1]. For all these reasons, several tens of thousands of tons of plastic are currently accumulated in this basin. Plastic is present in five environmental compartments: sea surface, water column, sea floor, coastline, and marine organisms [2]. Marine plastics originate from ship or land-based sources [3], the latter being of greater relevance [4]. A significant portion of the terrestrial plastic is transported to the sea by rivers. Previous studies assess that 80% or more ([5,6]) of marine plastic pollution comes from land through the rivers’ flow. Recent studies (e.g., [7]) at European level confirm that riverine systems act as converging pathways for discarded litter within drainage basins, and estimate that between 307 and 925 million litter items (plastic fraction represented 82%) are released annually from European rivers into the ocean. Rivers are able to transport plastic litter over long distances from inland to the coastal or lake environments, where they can interact with biota and the water ecosystems [8,9]. When plastic waste reaches the sea, it sinks in sedimentary sludge (70%), it remains in the water column (15%) or it ends up on beaches (15%) [2]. Marine litter (hereafter ML) transport by rivers is a crucial and complex component of the global plastic pollution challenge [10]. A large contribution of macrolitter inputs are from high-income countries, which suggests that the enhancement of waste management would not be enough to solve the ML pollution problem. An abrupt change in human consumption habits and behavior would also be necessary to curb waste generation at the source, and this requires regulating plastic production and use on a global scale [7]. It is estimated that environmental plastic pollution will more than triple by 2060 unless the current trend is reversed. [11]. To prevent this process, public policies have introduced several specific initiatives both at national and European level. The Marine Strategy Framework Directive [12] and the European Strategy for Plastics in a Circular Economy [13] are particularly relevant examples, amongst the others [14,15], the latter being focused on issues related to single-use items.

Studies on the transport and dispersion of natural particulate matter, sediment grains, macroalgae, wood and plants, pumice and planktonic organisms ([16,17,18,19], among the many) in open ocean and coastal areas have been ongoing for a long time. Meanwhile, considerable experience was achieved in terms of predicting the transport of dispersed materials at sea, both for oil spills and search and rescue activities (http://www.facebook.com/Toscaproject/ (accessed on 10 January 2023), [20,21,22,23,24,25,26]). These previous studies were particularly useful in contributing to an increase in the comprehension of the dynamics of transport of plastics at sea, but in recent years specific investigations of ML dispersion have been carried out [27,28,29,30,31,32,33,34,35]. When plastic debris reaches the sea, its slow degradation time and relatively high buoyancy facilitate long-distance transport from source areas [36]. As a result, the widespread distribution and fate of plastic in the sea can vary. Physical, chemical, and biological processes, together with typology and characteristics of the plastics, influence their transport towards the bottom, the offshore, or the coast [37]. Extreme events too, such as floods, tsunamis, and storms, can affect the flushing out to sea and transport of floating marine plastic debris, but only very few studies deal with this topic, at the present. Beaching of debris coming from Japan’s tsunami of 2011 onto shores of North America and Hawaii was studied by Carlton et al. [38], and the influence of the Arno Flood on the dynamics of ML deposition on the coasts of San Rossore marine park, near Pisa (Italy), was studied by Merlino et al. [39]. Equally, even if the number of studies and models of dispersion and distribution of macro- and microplastic in open waters has increased in recent years, there are still few reports in the literature concerning the beaching processes. Interacting factors such as bathymetry, coastline typology, presence or absence of sea and land breezes, waves, etc., concur in making the stranding processes complex. The dominant hydrodynamics in coastal waters controlling the transport of plastic objects in swash and surf zones differ significantly from the hydrodynamics occurring in the open ocean [40,41]. The restraining effect of a near-coastal boundary or shallow bottom topography may affect the drift response to local wind forcing [42]. The wave activity induces mass transport through Stokes drift as well as currents in the surf zone with parallel and perpendicular components to the coastline (longshore and cross-shore, respectively). In this latter direction, buoyant macroplastic particles are affected more by the onshore drift, so towards the shore, as they have a tendency to remain at the sea surface [43]. In this sense, attention has to be given in distinguishing movement and behavior of macro- and microplastic particles [42], and thus in estimating the average residence time (temporary retention in coastal areas before drifting offshore of both litter types [44]). More generally, particle-tracking models poorly represent the interactions of floating macrolitter (hereafter FML) with the shoreline. As a result, the numerical values for FML stranding, and those refloating, are unrealistically represented [45]. Finally, a great importance in the trajectory and fate of marine plastic debris is to be attributed to the physical characteristics and location of the litter sources, both in coastal [46] and in open ocean [47] modeling. All of this has great influence on the dynamics of plastic material accumulation on beaches [34]. Some beaches, if not cleaned regularly, literally appear to be covered with AMD, and this is especially true in autumn and winter, when the flow rate of watercourses increases and waves push floating objects towards the shore, induced by more frequent storms [39]. Isolated and/or protected areas within constructed zones suffer the most. In these areas beach cleaning is sporadic, as there are no facilities to arrange it, and the debris brought by the sea accumulates and stratifies through the years, often creating actual dumps with buried material. These can subsequently be dug up by erosion processes, and be returned to the sea at a later time [48].

The territory of Migliarino, San Rossore, Massacciuccoli Regional Natural Park (hereafter MSMRNP), located in the Italian western coast among the provinces of Livorno, Pisa, and Lucca, and affected by three watercourses (Arno, Serchio, and Morto rivers) is a perfect example of an area where the processes just described take place. The MSMRNP is the largest natural park in Tuscany, encompassed by a wide variety of environments of great ecological value with a total degree of protection and by inhabited areas designated for both agricultural and tourism uses, thus exerting significant anthropogenic pressure on the protected areas. Arno is the largest and most important river in Tuscany. It flows through densely populated areas and several important productive activities are located along its course, such as tanneries, shoe factories, textile industries, small- and medium-sized plastic products companies, etc. The Arno River flows just to the south of the protected marine area of the MSMRNP, and it continuously pours ML of all kinds into the receiving coastal waters, potentially affecting its beaches.

These reasons led us to choose this “A” protection zone of the park for our “study case”, whose aim is to describe the dispersal of floating plastic debris from the rivers toward the sea and the coastal areas surrounding the river mouth. In detail, the project, “A multidisciplinary method to study the Marine Litter Dispersion from the Arno River mouth: a study case to support citizen science” (ML-DAR Marine Litter Drifter from Arno River), funded by Istituto Nazionale di Geofisica e Vulcanologia (INGV) and carried on in collaboration with CNR-ISMAR and CNR-IFC, focuses on comprehending the dispersion dynamics of ML transported by the Arno River and discharged into the surrounding marine waters and areas. The study makes use of specific low-cost marine litter trackers (MLT), i.e., special drifters designed to simulate the buoyancy and shape of typical ML, and ocean numerical modeling.

The first step of the project was the construction of some MLT prototypes, their testing, and the implementation of a series of pilot dispersion experiments carried out in different conditions at the Arno mouth. Numerical models of ocean circulation and litter dispersion were used to predict the evolution of the MLT in the Tyrrhenian Sea waters and, at the same time, derived benefit from them in terms of data for calibration and validation. The first experiments involved only a few prototypes, while the next step of the project will be to produce a much larger number of MLT (if they will prove to be sufficiently suitable and efficient for our purposes) in order to carry out more robust and representative dispersion experiments focused on the Arno mouth in different scenarios of river flow and meteo-marine conditions. This will allow us to investigate as well the seasonality and impact of the waste contribution from the river. In this phase, wide support by citizen science is planned, both for the recovery of the objects thrown in the sea and, further on, for the construction of the final models of drifters, once those experimented are definitively tested and consolidated.

It is worthwhile to briefly summarize the types of drifters most commonly used, their specific use, and therefore the reason why we consider it appropriate to develop special devices for the type of monitoring we are interested in. In practice, it consists of a floating device equipped with a georeferenced tracking system and, possibly, instrumentation for chemical–physical or even biological monitoring of the hosting waters. Obviously, the drifter must be equipped with a data transmission system. Drifters have long been used for a variety of purposes, such as large-scale ocean current mapping, oil spill monitoring, and search and rescue operations. Depending on the different purposes, they may have different characteristics.

The most common research drifters are the SVP [49], the CODE [50] and the CARTHE drifters [51]. While following the near-surface (from about 1 to 10 m deep) flow, such devices are designed for studies of ocean circulation, plankton and runoff dispersion, and oil spill modeling, attempting to minimize Stokes drift and direct wind forcing, or windage. (Windage is the effect of wind on items with a freeboard, i.e., an area protruding from the water. It depends on the drag-area ratio of the drifter, which is defined by the cross-sectional area below the water surface divided by the cross-sectional area above the water line that is directly exposed to the air). Other types of drifters, such as ARGOSPHERE [52], are designed specifically for hydrocarbon spills, thus being driven by the wind and the currents within the first few centimeters of the water column. The usual communication method of these drifters is by satellite (IRIDIUM, Globalstar-SPOT, ARGOS). This is justified by the fact that they are designed to follow large-scale marine currents in real time, often very far from the coastline. Indeed, most drifters are used to investigate the transport of seawater (e.g., [53,54,55]) and ocean dynamics (e.g., [56,57]) to estimate mixing regimes and diffusivities from submesoscale and mesoscales to larger scales [58,59,60]. In these cases, they are used in a Lagrangian framework (see [37] for a recent review). Circulation features in the Mediterranean Sea have been studied through extensive Lagrangian drifter experiments (TOSCA project, http://www.facebook.com/Toscaproject/) (accessed on 10 January 2023) carried out in northern Aegean, near Toulon in France, near the Balearic Islands in Spain, and in the Gulfs of Naples, Trieste, and La Spezia in Italy [61,62], and in the middle and southern Italian Adriatic coasts [63]. Since the motion of drifters on the ocean surface is influenced by windage, Lagrangian drifters must be conceived to minimize wind-driven velocities and thus be representative of effective surface ocean currents [64,65,66]. Drifter designs that focus on transport and dispersion of pollutants such as floating marine debris or oil spills typically have a lower drag-area ratio and target the transport in the upper meter, also taking into account wind- and wave-induced motions, which primarily influence the transport of larger plastics [67]. Recent drifter designs have adjustable drag-area ratios to simulate different floating objects with different characteristics [68,69].

Given the focus of this study, the shape and buoyancy characteristics of the drifters must be as close as possible to those of the typical MLs discharged by rivers into the sea. Thus, standard drifters as those listed above cannot be considered ideal for this type of monitoring, mainly due to their inertia, exposure to the wind, and their material composition. In addition, the cost of such drifters is so high that it would make it extremely difficult to support an experiment based on several campaigns repeated at short intervals for at least a year. Indeed, a relatively high monitoring frequency is essential to study the flux of ML in different seasons and weather conditions. For these reasons, our strategy to guarantee sustainability and accuracy to our experiment was the adoption of ML itself as drifters, such as plastic bottles or small pieces of wood, which we equipped with the electronic devices necessary for remote tracking. Following the same principle of controlling costs and maintaining the main characteristics of the ML, we abandoned satellite communication (which is expensive and requires many batteries to sustain use) and opted for GSM and LoRa networks. This solution presents important advantages in terms of costs and energy consumption. Although in the Mediterranean Sea the ML carried by rivers usually remains close to the coasts, it can reach distances where the GSM or LoRa signal is undetectable (more than a few tens of kilometers from the coast). For this reason, we equipped the drifters’ electronics with a flash memory independent from the communication signal, on which all sampled position data are recorded at regular time intervals (together with data from any on-board sensors). As soon as the MLT recovers the GSM or LoRa connection, either because it has come close enough to the coast or because it has found an occasional bridge (i.e., ships), it will transmit the entire archived data, including the drifter positions in time. Since the disconnection interval may be considerable, even of the order of several weeks, it is essential that the drifter is equipped with adequate power autonomy in order to acquire all the data. The drifters were thus equipped with a small photovoltaic (PV) panel, about 5 by 5 inches, which is particularly efficient and suitable for marine uses. This PV panel has proven capable of generating more than 10 Wh on a typical winter day, and almost three times as much in summer—a more than adequate amount of energy for the on-board low-consumption electronics. Thus equipped, if (very rare) failures do not occur, our drifters have proven to have unlimited operational range.

After several campaigns of coastal monitoring of beached waste [39,70,71,72,73,74,75] we could classify ML into two macro categories based on (i) the characteristics of buoyancy and (ii) the exposure to wind. One category includes all types of plastic bottles, which have a relatively important emerged surface that is subject to the direct action of the wind, while the other category includes “wooden-board-like” waste, which stands semisubmerged when floating in water.

This article is organized as follows: the material and methods adopted are presented in Section 2. The main results are presented in Section 3, while the discussion and conclusions are offered in Section 4 and Section 5.

## 2. Materials and Methods

### 2.1. Study Area

The area selected for the ML-DAR experiment is the Arno River mouth (Figure 1).

This choice is based on the previous experience of some of the authors in SeaCleaner, a citizen science project [70,71], which involved more than 1000 students and citizens in monitoring beached litter during 2014–2016. Through the project, a total beach area of 32,154 m^2^ was surveyed and 34,027 litter items were collected and cataloged [72,73]. In particular, the project focused on five areas located in the coastal zone of the Pelagos Sanctuary, including MSMRNP. The monitoring results indicate a greater accumulation of waste in natural areas compared to urban and urbanized areas and an important role of rivers in the transport of these pollutants [72], confirmed by other similar studies performed in southern Tyrrhenian Areas [76]. In particular, on the beach of San Rossore, i.e., the area of total protection within the park (access not permitted except for research purposes), much waste coming from the Tuscany hinterland was found. The Arno River is an important Italian water course crossing the Tuscany region and flowing through large cities such as Florence and Pisa, and industrial and productive centers such as the province of Prato and Pontedera. The SeaCleaner monitoring project confirms that the composition of litter that reaches the beach is related to land uses (as indicated in [77]), socioeconomic activities (as indicated in [78]), and littering behavior (as indicated in [79]). The limitation to tourism in this area of the park guarantees that the sea is the only source of ML, so its transport and accumulation are related only to natural drivers. Particularly important roles are played, for instance, by meteo-marine conditions, such as wind, waves, and currents [80]; geomorphological characteristics, such as beach slope [81]; morphodynamic state [82]; level of sheltering [83]; and occurrence and frequency of extreme events [39].

### 2.2. Wind, Currents and Arno River Discharge

The marine circulation in the study area is dominated by a characteristic feature of the northern part of all semienclosed basins [84]. In the case of the present study domain, its name varies according to different authors: northern current (NC) in Millot [84], Liguro–Provençal–Catalan current (LPC) in Pinardi et al. [85], and Ligurian current (LC) in Millot [86]. We decided to call it, hereafter, LPC. As demonstrated in several studies, both using direct current measurements (e.g., [87,88,89]) and numerical modeling (e.g., [85]), the LPC, which ranges from the surface down to approximately 500 m [90], follows a cyclonic path parallel to the coast, ranging from the southern boundary to the western boundary of our study area, continuing its path towards the Catalan Sea in southwestern direction [90].

The LPC is generated in the eastern part of the Liguro–Provençal Basin (LPB), north of Corsica, by the merging of two northward currents, one flowing along the eastern coast of Corsica, the eastern Corsica current (ECC) and the other one flowing along the western coast of Corsica, the western Corsica current (WCC), as shown in several studies (e.g., [85,91]). After the merging of the two branches of Corsica current, the LPC flows offshore from the coast of Tuscany, approaching the coast in proximity to Liguria and then flowing cyclonically along the coasts of Italy, France, and Spain [92]. Figure 2 (left column) shows the seasonal surface circulation patterns for winter and summer computed from a reanalysis of the Mediterranean Sea, the MEDREA16 dataset [93,94] during a 20-year period (1999–2018).

As observable from Figure 2 (left column) the ECC reaches its maximum in winter, followed by a progressive decrease in its intensity up to summer, where a reversal of the surface portion of WCC (observed also in [95,96,97]) can be noticed with the formation of an anticyclonic structure that is referred to as Ligurian anticyclone in [96]. The WCC, weakening in summer, as highlighted in [98], is tightly correlated to the damping of wind stress in the area during the warm season (Figure 2, right column), when seasonal wind patterns for winter and summer computed from the ECMWF ERA5 atmospheric reanalysis ([99,100]) during a 20-year period (1999–2018) are shown. The wind pattern shows strong seasonal variability, both in terms of wind speed and prevailing wind direction, with an enhancement in wind energy during winter (Figure 2, right column, upper panel) with respect to the summer conditions (Figure 2, right column, lower panel).

As highlighted in [88,101], the winter season wind regime in the western part of the LPB is dominated by the cold and dry Mistral, which reaches the northern part of the Tyrrhenian Sea and the Ligurian Sea blowing from the north–northwestern direction through the Rhone Valley and the Gulf of Lion, and by the Tramontana northerly wind. The two winds interact with an additive effect due to their prevailing direction [102]. In the eastern part of the basin, along the coast of Tuscany, the prevailing wind during winter blows from the east in offshore direction and this air mass converges with that coming from the western part of the basin up to Cap Corse, in the northernmost part of Corsica Island (also observed in [101]), generating a southerly wind blowing towards the coast of Liguria. During summer, the southwestern part of the LPB is characterized by strong westerly winds, while intense winds blowing from the southwest direction can be observed in the central part of the basin.

In the eastern part of the basin, the wind regime shows opposite behavior with respect to winter, with the prevailing wind blowing towards the coast of Tuscany.

The Arno River discharge shows strong seasonality, with peaks in February–March and a minimum in July–August, as highlighted by Figure 3, which shows the monthly climatological discharge computed during the period 1999–2018 (time series and axis references in blue). Seasonal fluctuations in the Arno River discharge can affect the dispersion of marine litter throughout the year in the proximity of the river mouth, since fluctuations in the river flow result in modifications in the characteristics of the fluvial discharge-originated jet currents, which in turn interact in the coastal area with longshore currents, wind-driven currents, tidal currents, and density gradient currents, as well as strongly affected by variations in the freshwater input [103]. Figure 3 also shows the daily Arno River discharge (time series and axis references in red), highlighting peaks larger than 1000 m^3^/s in some events and a maximum overall discharge of approximately 1800 m^3^/s.

### 2.3. MLT Design and Construction: Base Requirements

In order to realize the MLTs for the first phase of the training test at sea, we thought to use already available material (bottles, containers, pieces of wood, etc.) rather than to design the support entirely from scratch. This is because our aim is to reproduce the behavior of the “real” marine litter in the sea through low cost and easy-to-replicate devices, so that more representative dispersion experiments involving a large number of them can be easily designed and implemented.

The selected items need to meet the following requirements:Be amongst those most frequently found in the top ten ML;Be suitable by size and shape to host a small solar cell, powerful enough to supply the electronic system mounted inside the floating object.

Regarding item “A”, the most recent data on macro ML categories found by monitoring 42 European rivers from 11 countries [7] show that 82% of the ML found was made of plastic, while 9% consisted of nonplastic items—mainly paper and cardboard—and finally only 4% consisted of metal. Totally, 8599 floating items were collected and analyzed.

Plastic waste consisted predominantly of fragments from larger objects and single-use items, such as bottles, tanks, small containers, cover/packaging, and bags. These results are in agreement with those found during the SeaCleaner in situ monitoring program performed on San Rossore beaches [73]. For what concerns item “B”, we agreed that liquid containers such as oil bottles, detergent bottles, or small tanks are the best candidates as it is possible to insert the electronic equipment and to seal them to prevent seawater from penetrating. Moreover, besides being among those included in item A, they often have more suitable shape and size, compared to smaller water bottles, to host a solar panel.

Another important requirement for the low-cost drifters was the cost-effectiveness of the data acquisition and transmission system. Here comes our decision to use the GSM standard, since satellite communication would have involved considerable complications in the design of the electronic device to be inserted, high additional costs [100], and energy consumption [104]. Since our target domain is coastal and we do not plan to launch these drifters in the open sea, we thought that the GSM transmission system would be a suitable low-cost choice. The drifters so equipped can remain connected to the GSM network mobile phone signals up to approximately 10–20 km from the coast; additionally, they can connect to nearby vessels and use them as a GSM bridge. Our drifters are also equipped with flash memory to record all acquired data, including GPS position. In this way, even if the GSM connection is lost for several days or weeks, all data are then sent, deferred, when the connection is reestablished. As a result of the various campaigns that we have carried out, we can say that all the drifters we have released into the sea from the mouth of the Arno River, even when they lost the GSM connection, were able to reconnect within a month at the most. Low power consumption enables the drifter to remain operational for an unlimited amount of time through a small solar panel of approximately 5 in × 5 in. This also facilitates possible recovery after beaching, for later reuse.

### 2.4. ML-Drifter Design and Construction, Electronic Devices, and Programming

Given the remarked importance of traceability, the insertion of the tracking device is a step to be followed carefully and it depends on the specific type of ML, as specified in the following paragraphs. Two tracking systems were tested in this pilot phase: one commercial and the other one built specifically for this application.

#### 2.4.1. Tracking Systems Description

As anticipated, we chose two different approaches for tracking systems: with commercial devices and with ad hoc realized tracking software. In the first case, we used the commercial device model TPK-905 manufactured by Winnes. It is a real-time satellite tracker with magnetic base, large-capacity battery (5000 mA), GSM connectivity, and USB charging connector. It can be managed through SMS, a dedicated application, or cloud platform. The operating mode was chosen with an hourly sampling to reduce the consumption of the electronic components of the device. The main limitation of this approach is the absence of a memory able to acquire and store the position in the absence of GSM connectivity, as the device has been produced for real-time data transmission. It is therefore an economical device, easy to mount on the marine litter (single block), and immediately ready to be used in coastal applications within the coverage distance of the GSM network. Since GSM and GPS antennas are part of the block, they have orientation limits. In order to guarantee the optimal operational and receiving conditions for the antennas, we placed the tracker, by means of specific spacers, on the highest emerged part of the device. To maintain low the center of mass of the device and favor its stability, we lightened the tracker by removing the magnetic base.

All data collected on the cloud platform can be viewed and exported in Excel format for subsequent processing procedure.

Concerning the second solution (ad hoc tracking software), we decided to develop customized software and hardware in order to overcome the limits imposed by GSM coverage. The MADUINO SIM808 card was therefore used: an ARDUINO ZERO card equipped with a GPS receiver, GSM connectivity, and an SD card slot. For this board, we developed specific software capable of transmitting real-time position data when the GSM connection is present, or to store them on the SD memory in the absence of GSM connection, ready to send them when the system reaches a GSM-covered area. In this way, we do not always have the data in real time, but we recover them at the first change in connection. This allows us to have the data even away from the coast where there is no GSM coverage, without using a satellite communication link, which is certainly much more expensive. This approach also allowed us to connect other sensors to acquire data concerning temperature, acceleration, etc., which would not be possible through commercial solutions. The GSM and GPS antennas are connected to the board by cables, which give us some degree of freedom to optimize their position. When necessary, ballasts (small weights) were used to have proper hydrodynamic stability of the ML so assembled. All data are collected on our server and therefore we have full control of the entire archiving procedure without depending on commercial services. As for the commercial tracking system, all data collected on the server platform can be visualized and exported in CSV format for subsequent processing procedure.

#### 2.4.2. Construction of MLT Supports

The drifters were made by using wooden tablets or plastic bottles/tanks as floating support. In the first case, the wooden tablets (size of about 38 cm × 38 cm × 4 cm) were treated with special wood stains to make them waterproof and then painted. Then, all the electronics were placed inside a watertight small box (about 10 cm × 10 cm × 7 cm) fixed on the upper side of the tablet. Moreover, to guarantee full waterproofing, the box was subjected to an additional resin sealing process or silicon applications. In the case of plastic containers (bottles/tanks), the insertion of the electronics was more complex. The upper part of the plastic object was cut off and the components were inserted, making sure to maintain the center of gravity as low as possible. If necessary, small ballasts were inserted to further lower the center of gravity. All these procedures were important to have the correct hydrodynamic balance of the MLTs, with about 50% of their body below the surface and ensuring stability during wave activity [105]. The recomposition of the bottle and the assembling of the anchor supports were done with resin or a special glue. The solar panel was attached to the major side of the MLT and connected to the box, all treated to make them watertight. Labels with the logo, QR code, and project reference contacts (previously printed and laminated) were also pasted on the free space of the devices. Each MLT also had an individual label with a sequential number. More details on the assembling and waterproofing of the components are available in Appendix A.

After adding electronics, batteries, and fixing the solar panels, we carried out tests in a water tank to ensure the correct buoyancy of the prototype drifters.

#### 2.4.3. MLT Data Management and Processing

Both tracking systems used were programmed to record the MLT position at regular time intervals, usually 60 min, but with the possibility of changing the recording frequency as needed. These data were immediately transferred to a data server on land, provided the tracker was connected to the GSM network, or stored in the tracker memory buffer and then sent once the connection to the GSM network was reestablished. MLT records consist of a list of events, each represented by the following attributes: date and time (UTC), geographic coordinates, speed of the drifter, direction of flow, GSM signal level, battery level, and information detected by the installed temperature sensors.

The accuracy of the GPS position is not recorded, assuming that the error is negligible on long routes at sea. The stored data are sorted in chronological order of the record events. This is necessary because the chronology in the drifter record would be altered when some of the data is temporarily stored in the tracker memory buffer.

Postprocessing of the data is also performed, to handle gaps due to failure to obtain GPS positions, to take care of chronological reordering, and for general quality control. In addition, the data are formatted to be read by GPX Studio software, so that routes can be easily visualized, speeds and traveled distances estimated, etc.

#### 2.4.4. Programming the Dispersion Experiments

Several MLT “launch tests” were planned and performed in different seasons and conditions (wind, weather, Arno flow rate, etc.) and with different numbers of objects at a time. The ML was differentiated by physical characteristics (i.e., shape, material) and tracking system at each launch except for the first one. Launch conditions ranged from deployment during the day to evening and night deployment. The planning and monitoring of every launch were supported by means of numerical circulation models coupled with Lagrangian particle-tracking models able to simulate and predict the evolution of the drifters at sea. The output of the models consists of the position of the particles representative of the ML at defined instants, including their potential beaching on the coast, thus providing information on the location and extent of the coastal areas affected by beached marine litter (BML). The experiments’ planning was based on the expected sea conditions derived from the predictions of the high-resolution model implemented in the present study for the coastal area and from the European Marine Forecast Service (CMEMS) for the offshore.

The detailed description of the modeling system is given in Section 2.4.5.

#### 2.4.5. Modeling the MLT Behavior

The Lagrangian component of the modeling system is based on the open-source framework OpenDrift [106]. The framework offers several Python-based modules designed for different particle-tracking purposes, including oil drift prediction, search and rescue, ichthyoplankton transport, macro- and microplastics drift prediction.

After preliminary tests performed using the PlastDrift module, the OceanDrift module was chosen as the particle-tracking model for this study.

The method used to integrate the advection equation is the Euler method; the wind_drift_factor parameter, which determines the fraction of wind speed at which particles are advected, was initially set equal to 0.02 and afterwards tuned through calibration with data acquired by drifters, as described in detail in Section 3.3; the chosen type of interaction of particles with coastline is stranding, which considers permanently beached the elements hitting the coastline by deactivating them.

The coastline used in OceanDrift simulations, which defines the potential stranding of a particle, is a satellite-derived product available through the Bathymetry Viewing and Download service of EMODnet Bathymetry (http://portal.emodnet-bathymetry.eu/) (accessed on 10 January 2023). Among the different available products, the coastline given for mean sea level (MSL) was chosen.

OceanDrift is forced by wind fields (zonal and meridional wind velocity) and by fields of surface ocean currents (zonal and meridional velocity). The wind forcing component is the ECMWF (European Centre for Medium-Range Weather Forecasts) forecast/analysis product while the ocean forcing component includes 15-min instantaneous-surface-current fields from the MEDSEA_ANALYSISFORECAST_PHY_006_013 Copernicus Marine Environment Monitoring Service product [107], and instantaneous-surface-current fields computed by a high resolution, tidal inclusive, implementation of the SHYFEM model (shallow water hydrodynamic finite element model [108]) in the coastal area, approximately between Cecina (LI) and Marina di Carrara (MS), as shown in Figure 4. A particular modeling effort was dedicated to the aforementioned area since it hosts the MSMRNP, which was the initial focus of the ML-DAR project.

SHYFEM is a 3D finite element hydrodynamic model solving the primitive equations under hydrostatic and Boussinesq approximations; the computational grid is an unstructured Arakawa B grid with triangular meshes [109,110].

The horizontal mesh for the study area (Figure 4) was built using GMSH software [111] and the horizontal resolution of the grid ranges from about 3.0 km offshore to approximately 200 m along the coastal strip of the study area. A particular grid refinement was applied in the areas from Viareggio to Cape Calafuria (just south of Livorno) and in the area surrounding the Secche della Meloria (just off the coast of Livorno).

The grid consists of 2800 nodes, 5290 elements, and 21 vertical z levels, with thickness variable from 1 to 10 m, covering a depth range from 1 to 90 m.

The bathymetry used was retrieved from the EMODnet Digital Bathymetry (DTM 2018, https://sextant.ifremer.fr/record/18ff0d48-b203-4a65-94a9-5fd8b0ec35f6/ (accessed on 16 March 2021), available through the Bathymetry Viewing and Download service of EMODnet Bathymetry.

Temperature, salinity, sea surface height (including tidal signal), and velocity fields are used as model initial and boundary conditions. In both cases, the variables used are derived from hourly mean fields from the MEDSEA_ANALYSISFORECAST_PHY_006_013 Copernicus Marine Environment Monitoring Service product [107].

Daily values for Arno River discharge are imposed. For hindcast experiments, daily values are derived from data recorded by the TOS01005191 automatic gauge station managed by the Regional Hydrological Service of Tuscany Region, located close to S. Giovanni alla Vena Valle, which is the most downstream gauging station providing daily discharge values for Arno River. For forecast experiments, daily values are derived from 1–10-day hydrological forecasts delivered by the Sveriges meteorologiska och hydrologiska institut (HYPE model area 9780931 [112]).

The river runoff is prescribed with constant salinity of 15 PSU, due to the lack of data in this regard, as also done in MEDSEA_ANALYSISFORECAST_PHY_006_013 product.

Atmospheric forcing is derived from ECMWF (European Centre for Medium-Range Weather Forecasts) analysis and forecast products at 1/8-degree horizontal resolution; the variables provided to SHYFEM model are zonal and meridional 10 m wind components, mean sea level atmospheric pressure, surface solar radiation downwards, 2 m air temperature, 2 m dew point temperature, total cloud cover, and total precipitation, which are used to estimate air–sea fluxes through bulk formulas.

A constant value of 2.5 × 10^−3^ for the wind drag coefficient and for the bottom drag coefficient was used. In this latter case, a quadratic bottom friction term is used in the momentum equations. An upwind scheme is used for both the horizontal and vertical advection in the transport and diffusion equation for scalars, with a diffusivity constant equal to 0.5 m^2^ s^−1^.

Horizontal eddy diffusivity is computed using the Smagorinsky formulation [113]. Vertical viscosities and diffusivities are computed by means of the GOTM k-ε model (General Ocean Turbulence Model, [114]). Astronomical tides are included through tidal potential and load tides.

#### 2.4.6. Citizen Science Contribution

Several citizen science contributions support this project:The involvement of 5 classes of high school Capellini-Sauro of La Spezia, Italy, in the scholastic PCTO (In Italian “Percorsi per le Competenze Trasversali e l’Orientamento’’, i.e., “Pathways for Transversal Skills and Orientation”), through National funding (INGV’s Project ML-CSA Study the Marine Litter dispersion: Citizen Science Application case) and EU/Horizon 2020 funding (ISMAR-CNR’s projects MARTA, BluesSchoolsMed, and NAUTILUS). High school contribution led to a general improvement in the design of the MLTs: the advanced knowledge of specific materials for the marine environment of the teachers and technicians involved in the project resulted in advancement in the choice of materials and in the assembly methodology of the two different types of floating structures selected (table and bottle/tank). Improvements and higher efficiency were also achieved in the electronic equipment, and the 3D design of a polyactic acid (PLA) case to host several components inside the watertight boxes was realized.During this PCTO, a considerable number of MLTs were built to be used for one or two final dispersion experiments, planned for the year 2022.Actions related to the release of drifter prototypes during the test phase, in 2021. In this period, from April to December 2021, we involved the State Police dive team (CNeS—Centro Nautico e Sommozzatori della Polizia di Stato) of La Spezia, which provided logistic support for the launch of the MLTs at Arno River mouth.As anticipated, a tag with a QR code and the inscription “If you catch me beached, contact us!” was affixed to the outside of the drifters, so that in the event of beaching, cessation of emission, etc., any persons who had found and collected the object could contact us (through e-mail or QR code) to provide information on the recovery of the drifter and to possibly turn it back. Lifeguards in the Marina di Pisa and Tirrenia areas, near the Arno mouth, were also alerted during the spring/summer period of 2021, when several test launches have been done.

## 3. Results

### 3.1. Technical Results

#### 3.1.1. MLT Prototypes

The upper panels of Figure 5 show a sketch of both the MLT prototypes, while the lower panels present some pictures of their realizations used during the ML-DAR pilot project (from April 2021 to December 2021).

The volumes of the bottle-shaped prototypes range from 2 to 5 liters. Such differences result in different emerging surface directly exposed to wind forcing. The tablet-shaped ones (square shape, side of 38 cm) have chamfered vertices to decrease possible damage to boats in case of an impact. At the beginning of the project, all relevant and interested organizations/parks/authorities were notified of the beginning of the activities.

#### 3.1.2. ML-Drifters’ Performance during Testing Phase

In Table 1, we report the data related to the launches of the different prototypes, carried out during the testing phase period, from April 2021 to December 2021.

The test phase has given excellent results in terms of durability of the prototypes (permanence in the sea up to 42 days, some drifters recovered and reused for more consecutive launches, see Table 1), and also of power-supply recharge via the photovoltaic panel. The MLT performance was very good for all the prototypes of the commercial tracking system, while we noted that the customized tracking system had problems during the launch of December. The customized tracking system needs more energy, and during test launches it emerged that one panel did not prove to have sufficient energy during winter seasons, even if, during the launch of December, we used a more efficient type of panel, designed for harsh marine environment applications compared to those used previously. As the system customized through MADUINO undoubtedly offers significant advantages in terms of trace completeness and temporal resolution, we opt for the use of such a system for the next prototypes, solving the problem of power supply providing the next MLT for winter use with two panels, when possible. Moreover, bypassing the height sensor on the sea level (unnecessary for our purposes), one can still greatly reduce the energy consumption.

### 3.2. Preliminary Results Obtained with MLT Prototypes Deployment

A total of fourteen drifters were deployed in front of Bocca d’Arno, Marina di Pisa, Italy, on five different dates from April 2021 to December 2021, with one, three, three, three and four MLTs at the same time. The deployment location was the same (43.68202° N, 10.26484° E, with uncertainty of 0.005° on both coordinates) for all launches, chosen to be in the coastal area at the end of the Arno plume. Information related to MLT deployments, displacement, and beaching is reported in Table 1.

The data set we obtained allowed us to follow the movement of “supposed” ML from a minimum of a few kilometers (launch of 9 August) to a maximum of 852 km (launch of 16 April). The lifetime of the tested MLTs ranged from a few hours (launch of 9 August) to 43 days (launch of 16 April). Excluding the two launches of August, in which MLTs were beached almost immediately after release, the mean daily distances covered by MLTs ranged from 16 to 24 km (Table 1). Figure 6 shows the complete trajectories of the MLT during each one of the launches performed during the testing phase period. Figure 7 and Figure 8 focus on the launches performed in September and December 2021. They are the most interesting ones because of the length of the trajectories and the contemporary presence of more drifters of different types. In these two pictures, the arrows illustrate the wind and current conditions for selected time intervals (in the zoom-windows), to give an overview of both the external forcing acting on the MLTs during these time periods. Examples in which beached objects refloated, probably due to winds and waves, are shown in Figure 7 and Figure 8.

The first test launch was on 16 April 2021 with a single tablet-shaped MLT. The commercial tracking system (WINNIES) did not return us the complete track, and there were time periods for which the lack of GSM connection did not allow us to reconstruct the entire trajectory followed. In any case, it definitely headed westward (i.e., offshore) then it began to move northward, following the main pattern of the LPC. The drifter arrived in front of Arenzano (lat = 44.404464°; lon = 8.687947°) on 3 May 2021, where it was carried only a few meters from the beach and then washed back out to open sea and arrived on 10 May 2021 in front of Cannes, and on 12 May 2021 in front of Saint Tropez (both French locations). The loss of connection at this stage did not allow us to trace its precise trajectory, but it was probably advected for 12 days by the LPC, which presents a cyclonic pattern, and reached the French coastal zone first, and then the southeast Ligurian Gulf, in Italian territorial waters. Indeed, the signal was again detected on 24 May in front of the Portofino promontory, and then it started to follow the current again, being transported westward in a clockwise turn, once more. On 27 May, the drifter beached near the breakwater of the port of Savona (lat = 44.30593° N; lon = 8.49224° E). After approximately two days (on 20 May) it was flushed back into the sea, and on 28 May it finally stranded on the beach of Spotorno-Bergeggi (lat = 44.23672°N; lon = 8.43487°E) at 20:00. Unfortunately, we presume that it was collected the following morning during beach cleaning and thrown into a garbage can, so we could not obtain any feedback through the channels we established. Albeit this lack of communication was a bit upsetting, a trashcan was the appropriate destination for the end of the life of an AMD.

On 9 August 2021, three MLTs with three different shapes were launched at the same time: a table-shaped one (like the one of April) and two bottle-shaped with two different sizes: a detergent tank of three liters and an oil tank of five liters. In this case, the drifters’ travel was very short: the sustained southwesterly wind (daily breeze) quickly pushed all of them towards the coast of the MSMRNP, where they beached very close to each other. We could then rapidly recover them.

A few days later, on 12 August 2021, a new launch and a similar fate, but in this case the wind came from the northwest direction. Even in this case, the trajectories were very similar for the 3 MLTs, mainly influenced by local coastal wind forcing. In both the August cases, the result is a quick beaching of the objects just as they came out of the river plume.

The launch of 15 September 2021 is the only one when the release was done late in the evening (7 p.m.) rather than in the morning. We opted for this choice in order to avoid the dominant day breeze regime typical of the season, which would immediately push the drifters back onto the beach. In this way, we also simulated the conditions that characterize the evening/night period, with the reversal of the breeze, and we could investigate its influence on the transport of ML from the river to the sea. The two bottle-shaped MLTs (with two different volumes) and the table-shaped one started to follow different trajectories from the second day at sea (Figure 7). Under the same sea and wind conditions, the tablet followed a more “indecisive” trajectory, lagging behind the two bottles on which the effect of windage prevailed, due to the larger emerged surface. The five-liter MLT always traveled faster than the two-liter MLT and the tablet MLT, always following behind. On 18 September, the five-liter bottle reached the inside of the Manarola port. At night of the same day, it headed again offshore, followed by the two-liter bottle and by the tablet MTL under conditions of weak to absent coastal currents and dominating winds (breeze regime with wind toward the coast during the day, and toward the sea in the evening). The situation changed the day after, and a strong swell and sustained winds pushed the two bottle-shaped MLTs toward shore, where they finally beached, one in “Fossola” and the other in Martolga Bay (between Levanto and Monterosso). The table-shaped MLT, however, remained relatively offshore, opposite to Forte dei Marmi, always set back from the other two. At the end of the swell, when the breeze regime set in again and the evening wind was blowing from land, the five-liter MLT was pushed out at sea, although shortly afterward it beached again in Levanto, where it stopped transmitting. The two-liter MLT was subsequently recovered thanks to the staff of Cinque Terre Park on Fossola beach, while the tablet-shaped MLT, less influenced by coastal breezes and winds, was captured by the Ligurian current and arrived once again in the area of Saint Tropez on early October. Here, the drifter was beached and it was taken to the landfill following the morning beach cleanups, as in the case of April.

In the last launch (16 December 2021, Figure 8) the paths of the two bottle-shaped MLTs and those of the tablet-shaped ones were initially very similar, but within one day the two bottle-shaped MLTs were traveling faster, as already happened in the launch of September. The tablet-shaped MLT also showed a slightly different trajectory in the following days compared to the other two MLTs, which behaved very similarly to each other. Starting from 18 December and for a period of three days, the three drifters were trapped in an eddy in front of Forte dei Marmi, and even here the table-shaped trajectory deviated a bit from the other two. During the following days, the two bottle-shaped MLTs presented similar behaviors, probably due to a similar response to the direct wind forcing, and they were always slightly ahead of the tablet-shaped drifter. Finally, the three drifters were beached during a swell with wind of about 11–16 knot (F4 in Beaufort scale) and very weak current (data from Consorzio Lamma and Copernicus Marine Service), in three different locations between Sestri Levante and Pieve Ligure, on 24 December.

The fourth drifter (tablet-shaped), which also beached during that swell, then departed the following day. Unfortunately, due to charging system malfunction, it did not send its position again in the following days until 3 January 2022, when it sent its last signal in front of the Hyeres Islands, France (lat = 42.86023° N, lon = 6.33283° E).

Overall, considering all the launches carried out during 2021, there were several episodes of refloating following stranding on the beach.

### 3.3. Testing Phase’ Forecast and Models

The coupled modeling system was used since the first deployment in April 2021 in forecast mode in order to support the planning of the experiments and to provide information on the most probable trajectories and potential beaching locations. In the beginning, the ocean data driving the dispersal model were those of the analysis and forecast from the Copernicus Marine Environment Monitoring Service (CMEMS). Then, owing to the inadequacy of the spatial resolution of the circulation model (approximately 4.5 km) for a coastal targeted area—the protected area of MSMRNP, which was in the very beginning the real focus of the study—the high-resolution model based on SHYFEM introduced in Section 2.4.5 was implemented. The Lagrangian system runs on high resolution currents when available and it automatically switches to low resolution when the particles exit the high-resolution model domain. In this way, we are able to capture very coastal- and river-driven dynamics close to shore, whereas we rely on the European Service in the offshore where the influence of the coast is limited.

After the experiments at sea were terminated, the positions recorded in real time by the drifters were used to calibrate and validate the Lagrangian dispersal model. One of the most relevant factors of calibration for the particle-tracking model is the wind drift factor (WDF) [115], which represents the magnitude of wind-driven surface current directly induced by wind stress. Such a factor is usually confined between 0.01 and 0.06, meaning that the intensity of such currents can be approximated as 1% to 6% of the intensity of the 10-m wind. We then varied the WDF from 0.00 to 0.06 through bins of 0.005 for the deployments at sea (see Table 1), and we simulated the dispersal of 200 particles for each class of WDF. The particles were released at the same time as the physical drifters within a radius of 1 km from the point of release at sea. The radius of release and the most representative number of particles to release are as well sensitive parameters of calibration, which we will address in future studies of litter plastic dispersal in order to further improve the modeling system. The experiments conducted in August 2021 were considered nonrelevant for calibration since all drifters were beached within 3–4 h from the deployment and the differences related to different WDF in the Lagrangian model would not be appreciable. The calibration configurations were first assessed in terms of root mean square error (RMSE) based on the separation distance between the drifters and the cloud of simulated particles at each available pair of longitude–latitude registered in time. The RMSE is calculated as follows:(1)$$RMSE\left(t_{i}\right)=\sqrt{\frac{\sum_{n=1}^{N}dist_{i}{\left(x_{m}\left(t_{i}\right),\;x_{o}\left(t_{i}\right)\right)}^{2}}{N}},$$
where *N* is the number of simulated particles (200 for each WDF class, in our case), *dist_i_* is the distance at the instant *t_i_* between the real drifter position (*x_o_*), and the *i*th simulated particle position (*x_m_*). However, the meaning of a certain value of RMSE is quite different if such error is registered after one day of simulation or after longer periods. To account for the different model skills in relation with the length of the simulation, we further introduced another metric for the evaluation of the dispersion model accuracy. Following [116,117] we calculated a nondimensional index (*s*) assessing the model quality, defined as the mean of the separation distances of the simulated particles from the observed drifters, normalized by the lengths of the observed trajectories. The index is then time-dependent and it can be calculated as follows:(2)$$\left(t_{i}\right)=\frac{1}{N}\overset{N}{\underset{n=1}{\sum }}\frac{\sum_{t=t_{0}}^{t_{i}}dist_{i}\left(x_{m}\left(t\right),\;x_{o}\left(t\right)\right)}{\sum_{t=t_{0}}^{t_{i}}l_{oi}\left(x_{o}\left(t_{0}\right),\;x_{o}\left(t\right)\right)}$$
where *l_oi_* is the distance traveled by the drifter from the beginning until the instant *t_i_* while the other parameters were previously introduced. Moreover, we propose an additional weight applied to the distance *dist* in order to take into account the number of beached particles with respect to the entire population of each category of WDF (200, in our case). The weight penalizes the mean distance of the particles in each class from the observed position of the drifter, up to a maximum of 20%. It is a multiplying coefficient of the distance *dist_i_* whose values varies from 1, if none of the particles are beached in the considered instant, to 1.20 if all the particles are beached. In this way, we penalize the classes of WDF that present the lower number of particles still active when the drifter is not yet stranded.

Finally, the skill score of the model can be calculated starting from *s*, as follows [116,117]:(3)$$ss\left(t_{i}\right)=\left\{\begin{array}{c} 1-\frac{s\left(t_{i}\right)}{n}\;\;\;\;\;\;\;\;\;\left(s\leq n\right)\\ \;\;0\;\;\;\;\;\;\;\;\;\;\;\;\;\;\;\;\left(s>n\right)\;\; \end{array}\right..$$

The parameter *n* is a tolerance threshold [116], which we considered equal to 1. The skill score values range between 0 and 1, with 1 being the score of models able to perfectly reproduce the observed trajectories. The skill score is 0 for models with no skill. Table 2 summarizes the result of the calibration experiments carried out for the deployments of April and December 2021. Bold values of WDF represent those which minimize the mean error of the model and maximize its skill score. Unfortunately, only these two deployments were useful for the model calibration/validation. Indeed, the launches of August 2021 were too short to be representative (all of the order of a few hours, as anticipated), while those of September 2021 would have been suitable in terms of observations, but in all cases the modeled particles were unable to track the observed trajectory for a representative time, being beached too early regardless of the WDF, so we decided to discard these experiments. The analysis of calibration metrics suggests that there is not a clear distinction between tablet-shaped and bottle-shaped behavior of the modeled drifters. The value we chose for the time being is 3.00% without distinction between bottles and tablets, and it will be used for the next experiments in forecast mode to predict the drifters’ trajectories and support their recovery and experiment planning. Then, once the future real-time experiments are concluded, we will use the recorded positions to further increase the drifter population and so improve the statistical significance of the experiments. This iterative methodology will be applied for every deployment in the future, so the model will recursively improve. We are, however, well aware that due to the small number of launches performed so far and to the limited number of drifters involved at each experiment, the actual calibration is not particularly robust and we believe that a much higher number of deployments in different meteo-marine conditions will allow us to differentiate WDFs according to the shape of the drifters. This enlightens both the difficulty and the importance of having relevant and conspicuous information on floating objects trajectories at sea for model calibration and validation purposes. In the meantime, this proves the great importance of low-cost drifters, relatively easy to assemble and to use.

Figure 9 shows the April deployment in maps of dispersion of the cloud of modeled particles (colored dots) compared to the observed position of the tablet-shaped drifter (black cross). Each panel is representative of the modeled and observed positions in a different instant after deployment. The instants shown are approximately 3 and 12 h, 6, 12, 15, 19, 22, and 24 days, after deployment, from top to bottom and from left to right, respectively. The modeled particles are gathered by groups of 200 items, and the colors represent different values of WDF. It is extremely important to see how the modeling system is able to give reliable information on the drifter position even after a very long interval from deployment. It is to be noted that, unfortunately, the observation record is not continuous: several holes are present in the tracking time series, sometimes even particularly long. As an example, we have a five-day gap from 17 April to 21 April and a five-day gap from 23 April to 27 April. Again, this helps understand the importance of consistent, long, and reliable tracking records.

The simulations of all deployments of April, September and December 2021 are included in the Appendix A as animations. 

## 4. Discussion

Our experimentation with low-cost drifters, intended for tracking ML in coastal areas, has given positive results in terms of performance for prototype devices, preliminary data collected through their tracking and modeling performance and reliability. Indeed, the experiments carried out so far were helpful to depict a preliminary interpretation of the behavior of the waste coming out of the Arno River in different weather conditions and in different seasons.

Our customized MADUINO-based tracking system equipped only with GSM communication proved to be particularly fit to overcome the problem of potential lack of coverage, while remaining in areas sufficiently close to the coast. Positioning and data transmission analysis have shown that geolocalized data provided are accurate and present high temporal frequency and very minimal gaps. Such requirements are necessary to investigate submesoscale processes. Moreover, the MLT deployments of September suggest that our devices can also operate during stormy weather conditions with wind speeds up to 28 Km/h (16 knots) and wave height up to 1 m. Owing to our approach, based on the use of realistic ML selected amongst the objects commonly floating at sea or beached on shores, it will be possible to extend the typologies of low-cost drifters including new shapes and dimensions, and to study and classify their behavior according to different areas (coast, open ocean, estuaries, etc.) and environmental conditions. In addition, the adaptability of our electronic devices to different types of support makes this kind of drifter suitable to more standard uses, providing drogues, for example, in the tablet-shaped type. Lagrangian drifter experiments with different drag-area ratios can provide fundamental insights into the dynamics of floating AMDs of different shapes at the sea surface. They are very helpful for calibration and validation of particle-tracking models [34,42], and they can be crucial for understanding the different effect of wind, waves, and currents on the floating objects with respect to their characteristics and shapes, resulting in different buoyancy ratios and response to wind drag [42,118] and current advection, thus ultimately influencing their dispersion [119], distribution, and residence time in the seawater or beaching and reflow events [120]. Other previous studies highlight how the use of low-cost, mobile phone-based drifters, easily implemented and adapted to different environmental conditions, can be a good solution for experiments in coastal areas ([104,121,122,123,124] among many others). For the time being, though, we have not found among them any type of drifter equipped with all the features that our MLTs present: customized tracking system with data logging, adaptability to different types of media, and autonomous charging system, all integrated.

For the specific ML transport and dispersion study, designed drifters should have specific characteristics concerning buoyancy and windage, different from those of classical Lagrangian drifters. In a recent very interesting experiment, drifters have been designed to simulate a plastic bottle [105]. The drifters were launched into both the sea and the Ganges River, and their transport was studied. This scenario is the closest one to our study case that we are aware of in literature. However, unlike our system, those drifters did not have an autonomous recharging system and the “bottle-drifters” were objects built ad hoc, and not real “trash objects” on which a tracking system has been mounted, as it is in our case.

The different MLT models we realized and tested are targeted at the motion processes of floating items in the surface layer (0.5 m), where wind- and wave-induced motions have an important impact on the transport pattern of floating AMD [67]. They have been able to represent the buoyancy and the behavior of ML (of a certain size) beaching and refloating processes. These are challenging issues, influencing the transport pattern and stranding locations of AMD, with consequences that affect the dynamics of accumulation and the formation of standing stocks of AMD [39]. Poorly represented in most ML transport models [125], they depend on several mutual interacting factors: the buoyancy of the objects [120], the structure of the coastline, bathymetry, winds, waves, and tides [125]. These factors have greater influence in northern seas, while in Mediterranean cases they have little importance compared to other factors, particularly to the wind, as the Mediterranean Sea circulation is strongly wind-driven. Therefore, the results of our experiments could be useful for wind- and wave-related parameters of calibration in particle-tracking models.

The complex interaction of winds, waves, and coastline morphology in the studied area is suggested also by the trajectories represented in Figure 6. Beaching and refloating cases are reported in the Results section, and highlighted in Figure 7 and Figure 8, where there is also evidence that floating objects, under low-wind conditions, tend to accumulate in submesoscale recirculation cells, which can lead to a prolonged residence time of floating AML in this part of the North Tyrrhenian coastline. The residence times of some days, experimented by more than one MLT in the coastal area between La Spezia and Viareggio, during the September and especially December launches, support these assumptions (Figure 7 and Figure 8). It is important to note that our modeling system was capable of predicting the deployment of December 2021, again corroborating these findings. On the contrary, the forecast (and simulation) of the deployments of September 2021 were not well reconstructed by the models. In this case, the virtual particles of all WDF tested beached too early—approximately by three days—and the beaching were located too far east—approximately by 20–30 km—compared to the location where the bottle-shaped drifters were actually found. Moreover, the tablet-shaped drifter was dispersed at sea for 20 additional days.

Observations by Suaria et al. [126] have shown higher abundance of floating macro-AMDs in the Tyrrhenian Sea, with respect to the Adriatic and Ionian Seas, with a large number of items in the Northern Tyrrhenian area. These local abundances were confirmed two years later [127] from values of collected floating microplastic: these ones result to be particularly abundant in the Ligurian Sea area, with density reaching 10.000 particles per Km^2^. Longer floating periods can also induce strong buoyancy reduction due to marine fouling and finally result in litter sinking and potential deposition on the bottom, or again they can be pushed to the shore with winds and Stokes transport, further contributing in the object’s beaching [128].

The observations of the MLT trajectories we obtained show that the transport of floating objects at the sea surface is strongly affected by the local wind field as well as related to their shape. During two out of the five test launches we carried out so far, local wind conditions pushed MLTs out of the river plume onto the beach within a few hours (launch of August 9 and 12, Table 1). In the other three launches, the drifters took the north (September 2021) and northwest (April and December 2021) directions, driven by the combination of wind and current. This is indeed corroborated by the ocean circulation model outcomes and by the wind observations at the port of Livorno (located approximately 20 km south of the Arno mouth), delivered by the Italian National Mareographic Network (Rete Mareografica Nazionale; https://www.mareografico.it/ (accessed on 11 January 2023) managed by ISPRA (Istituto Superiore per la Protezione e la Ricerca Ambientale; https://www.isprambiente.gov.it (accessed on 10 January 2023). The model-simulated sea velocities in the area around the river mouth were westward during April and December and northward in September 2021, with intensities ranging between 10 and 15 cm/s. On the same dates and at the same location, the measured wind speed varied between 2 and 4 m/s in the SSW direction in December and predominantly WSW in April, while in September 2021 the predominant direction and speed were north and 2–4 m/s, respectively. Being mostly driven by surface currents, the trajectories of the drifters showed similar behavior during the first hours, but then the movements of the objects began to differentiate, probably owing to the different influence of wind and current due to their different shapes, volumes, and buoyancy: for bottle MLTs, which have more exposed surface and greater buoyancy than tablet MLTs, the wind effect seems to predominate. This emerges more clearly from a comparative analysis of the trajectories of the two launches in September and December (Figure 7 and Figure 8). In both experiments, the table-shaped MLT always lagged behind the other two, sometimes recirculating in specific limited areas for some days (see the green trajectory in the box A of Figure 8).

In the launch of September, where the bottle-shaped drifters were of different volume (five and three liters), the five-liter MLT traveled faster than the one of three-liter, probably due to the effect of the wind (ranging between 4 and 7.5 m/s) on the wider emerged surface in the direction of motion (north), according to the results from the Copernicus Marine product “Global Ocean Hourly Sea Surface Wind and Stress from Scatterometer and Model” (WIND_GLO_PHY_L4_NRT_012_004, doi: 10.48670/moi-00305).

In the deployments of December, the trajectories of the two bottle-shaped (same shape and volume) were spatially and temporally very similar, until they approached the coastal area in front of Sestri Levante, Chiavari, and Rapallo (see box A of Figure 8), after approximately 8 days of floating. Here, the intense and suddenly reversed-to-coast action of the wind (almost 8 m/s towards northeast) on 24 December pushed the drifters on shore, causing their beaching, albeit the sea surface current was still heading to the northwest with relevant intensity (approximately 20 cm/s). The two tablet-shaped objects followed similar trajectories, but some differences are noticeable, such as the time delay with respect to the bottle-shaped (Figure 8). In fact, they both were pushed toward the shore as well by the intense and reversed wind of 24 December, but its action was not strong enough, probably due to their flatter shape, to strand them both to the coast. Indeed, only one was beached while the other one kept floating, eventually reaching French waters.

It is interesting to note that in all three launches in which the MLTs was initially entrapped in the LPC towards the north, at least one of the drifters followed the cyclonic pattern of that current and eventually reached French waters. In all cases such MLTs were table-shaped. None of the bottle-shaped MLTs deployed ever traveled farther than 140 kilometers from the point of release.

The understanding of the prevailing dynamics driving the different MLTs at sea and the knowledge of the main wind and current regimes suggest that important amounts of ML coming out of the Arno River mouth are not dispersed at sea for long distances, and they are often washed on shore within a few tens of kilometers from the river’s estuary, mainly during spring and summer. In these seasons, indeed, prevailing winds are westerly and present modest intensity. Thus, except during specific and intense events, the typical breeze regime is dominating local wind conditions. This implies a significantly longer interval of the daytime when local breezes blow toward the coast and the opposite at night, which covers a much shorter number of hours in summer. In this period, the prevailing current presents more variability than during winter and it often reverses towards the south. The combination of these wind and sea conditions favors the frequent beaching of litter in the coastal areas surrounding the river estuary in spring and especially in summer. The stranding events to the north of the river’s mouth largely interest the natural and protected area of San Rossore, where a significant amount of litter accumulates, favored by meteo-marine conditions and by the lack of coast management and litter collection, since the area is forbidden to public entrance. Important amounts of litter coming out of the river are as well transported to the south, impacting on the Port of Pisa and on the shorelines immediately to the south.

During autumn and winter, the LPC is well defined towards north, the prevailing winds are easterly, and the sea breeze regime is absent. Under these conditions, the litter of Arno origin often tends to be advected towards the open sea (as it was during our deployments in December 2021) and early stranding in the local shores are likely to be less probable under normal conditions. On the contrary, in stormy autumn–winter conditions, the river flow often presents important peaks inducing an enhanced input of litter to the coastal waters, which may soon be washed on shore in the case of sea storms and consistent wave action. Conversely, in summer such stranding events would preferably interest the coastal areas north of the river mouth, including the MSMRNP.

In this study, albeit the modeling system was not the main focus but rather a complement to the field activities, it proved to be an important source of information to plan our campaigns and to corroborate the hypotheses of interpretation of the observed litter behavior. In the experiments of December and April, the reproduction capability of the dispersion model was relevant, not only within the first hours from the deployments but also after several weeks. However, much more work is needed to make predictions reliable in all the ranges of different meteo-marine conditions, which is not the case for the time being, as suggested by the model misrepresentation of the litter trajectories for the experiment of September 2021. More processes should be included, such as the direct effect of waves (as Stoke transport), and more diversification of the parameters should be implemented and understood, such as the variability of WDF as a function of wind intensity and of typology of ML.

From the perspective of modeling, which often lacks observed tracking data for calibration and validation, the support of low-cost devices is particularly important. In our case, besides the community of research, their implementation is also extendable to a broader community of stakeholders and/or specialized companies. This opens vast possibilities to use this approach in the context of citizen science, with possible benefits for both researchers and the citizens involved. This is the natural approach of the big data paradigm, where the involvement of every user/citizen favors a dramatic increase in measurements. In our case, the many launches in different seasons, in a wide range of weather conditions, will allow an evident improvement of the quality and reliability of Lagrangian tracking models. The citizen science approach should thus ultimately lead to a better understanding of the mechanisms of plastic feeding and dispersal in the sea and, at the same time, increase the awareness of the issues faced.

Citizen science approaches to tracking waste dispersal have been used before, but mostly through the “passive” mode of “drifter recapture” [34,70]. A similar approach to the one we propose has been adopted only recently, using open-source technologies to track the movement of plastic bottle-shaped devices during an experiment conducted along the Ganges River and the coastal area of the Bay of Bengal [105]. In that study as well, the attention was given to both the scientific research aspect and the involvement of civil society in monitoring waste disposal and raising awareness of the problem. The method we propose here intends to strengthen the idea already proposed by Duncan et al. [106], for allowing citizens to build ML tracking devices, simplifying the technical aspects involved in their construction, and, most importantly, providing the tools to turn junk-objects of different shapes and materials into ML trackers, instead of building them ad hoc. This will enable people with minimal electronics/computer science expertise to assemble and use these trackers in real dispersal experiments, bridging the gap between data retrieval and modeling. Moreover, the use of tracking technology to follow AMD dispersion at sea jointly with mapping tools and with marine litter monitoring actions on the coast could provide a better understanding of the trash input mechanisms into the aquatic and marine system, pinpointing potential hotspot zones. The role of citizen science can thus extend from the standard contribution given by beach surveys (resulting in the collection of useful data on the amount and distribution of marine litter) to a more active role of building and deploying tracking devices that can provide useful data on the movement of litter at sea. The data thus collected represent great environmental and social value, creating a direct and effective link between scientific information for government policies and regulations on waste management and cleanup [129,130,131]. Open-source data collection in support of conservation is a key goal of the Decade of Ocean Sciences for Sustainable Development (2021–2030) [132].

From an educational and outreach perspective, these activities are indeed effective in raising awareness of the problem being addressed [133], and there is evidence that social marketing using these tools can foster lasting engagement and behavior change [134]. This is exactly what happened in the experience described in this paper, in which we involved a high school in both the design and construction process of our ML trackers. Among other educational and training benefits, the project enabled the school involved to be part of the European Blues School Consortium (https://decenniodelmare.it/ocean-literacy-blue-schools-network/) (accessed on 11 January 2023) and to participate in the BluesSchoolMed Erasmus + project (https://www.blueschoolsmed.eu, accessed on 11 January 2023). The commitment and interest shown by the four classes involved has led to important awards, including the first prize from the Liguria region for the best “Pathway for Transversal Skills and Guidance” in 2022.

Finally, in addition to marine litter tracking, our devices can be easily adapted for circulation and dispersion studies in a wide range of aquatic environments where Lagrangian measurements are scarce, such as the coastal zone, small and medium-sized lakes, rivers, and estuaries; that is, environments where the full capabilities of the most sophisticated drifters nowadays used are not needed.

The drag-area ratio can be adjusted by equipping the drifters with a subsurface drogue, thus enabling them to provide valuable information on transport triggered by subsurface currents. Moreover, other sensors (temperature, etc.) can be easily installed.

Our results highlight the potential for this consumer technology, hardware, and open-source software to become part of a low-cost integrated approach for oceanographic modeling of AMD motions in the seas. GSM transmission (or other standard mobile network technologies) is useful for reducing overall costs and allowing more data to be transmitted from a device, owing to the availability of bandwidth compared to satellite payloads. Cost remains a major limitation of tracking technology, hence the need to move towards cheaper, open-source software [135]. The rapid progress of new technologies represents a frontier that will allow the oceanographic community to explore more comprehensively and accurately the Mediterranean Sea, and more generally the world ocean in the near future.

## 5. Conclusions

In the present study, we successfully developed a proof-of-concept method using open-source tracking technology to help understand the transport of anthropogenic waste through aquatic systems (both riverine and marine). This work demonstrates a key step forward by moving from observations to tracking movement in “real time”. The MLTs were in fact built to replicate true movement patterns of floating ML (e.g., shape, size, buoyancy) by using real AMD.

Data obtained with preliminary dispersion experiments, using the first MLT prototypes jointly with the development of high-resolution particle-tracking models, highlight that the combined effect of winds and surface currents acts in a different way for differently shaped MLT; for the bottle-shaped ones, for example, a larger windage resulted in frequent beaching after a short path. On the contrary, tablets were found to be less affected by abrupt changes in direction related to wind and consequent sudden wind-induced beaching. As a result, the tablet-shaped MLTs were able to follow the main course of the LPC cyclonic gyre to the front of the Ligurian coast and eventually reach the sea and the French shores.

The adoption of open-source hardware and software potentially allows other interested people (community of the so called “makers” (nonspecialized people, i.e., not necessarily technicians/engineers, who use popular hardware/software, such as Arduino, to make a wide variety of tool/gadgets themselves; this community is becoming very large, as evidenced by the growing number, and success, of dedicated fairs), schools, associations, etc.) to freely replicate, modify, and share their experiences, which will hopefully improve the MLT design. For this purpose, a dedicated institutional website will be established that will also provide centralized data collection.

The current Marine Litter Drifter project from the Arno River can be extended to the outlets of other rivers, whose surroundings are also areas of AMD accumulation. In addition, it could be used in protected parks or other vulnerable areas (using low-cost MLTs and numerical simulations), which could provide valuable insight into predicting the dynamics and accumulation of AMDs. Such information would support both the organization of coastal cleanup interventions and the evaluation of policies for better environmental protection.

## Figures and Tables

**Figure 1 sensors-23-00935-f001:**
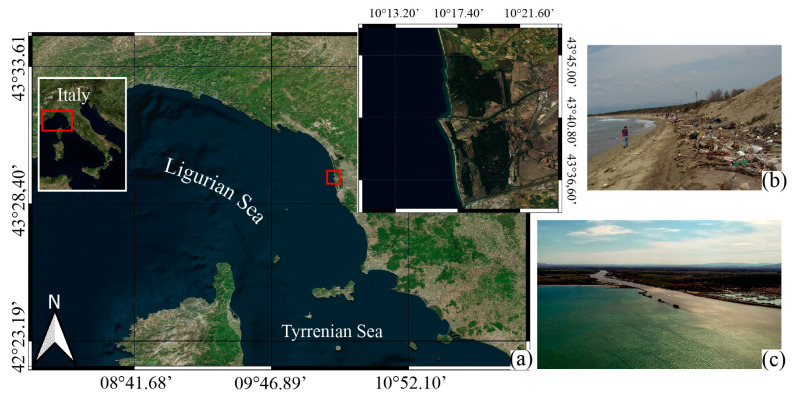
(**a**) The area in the zoomed square (red box in map) is the coast surrounding the Arno mouth. The shoreline of the MSMRNP panel (**b**) is located just north of the river’s mouth panel (**c**).

**Figure 2 sensors-23-00935-f002:**
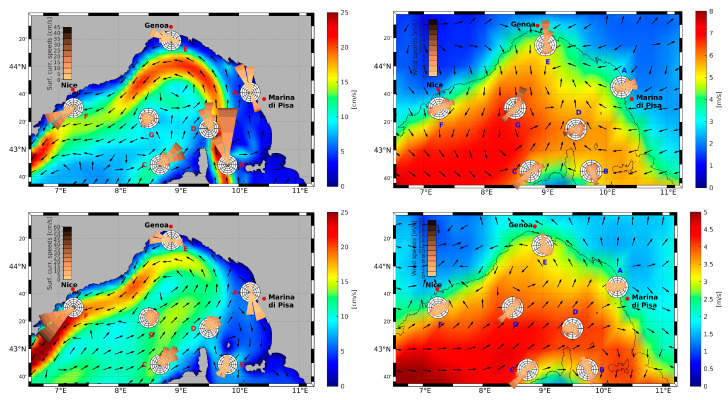
Left column: rose plots of seasonal surface currents speeds and directions (towards) superimposed to seasonal surface circulation pattern (upper panel: winter, lower panel: summer). Right column: rose plots of seasonal wind speeds and directions (from) superimposed to seasonal wind pattern (upper panel: winter, lower panel: summer). Concentric rings in rose plots represent the direction distribution with values ranging from 2% to 10%, with a 2% interval between each other. Cardinal directions are represented by 16 radiating spokes. Please note that different color bars are used in the panels. Labels from A to G define the following locations or surface circulation features. A: Arno River outlet area; B: eastern Corsica current (ECC); C: western Corsica current (WCC); D: merging of the ECC and WCC; E: northernmost portion of the Liguro–Provençal–Catalan current (LPC) in the Liguro–Provençal basin (LPB); F: easternmost portion of the LPC in the LPB; G: approximate center of the LPB.

**Figure 3 sensors-23-00935-f003:**
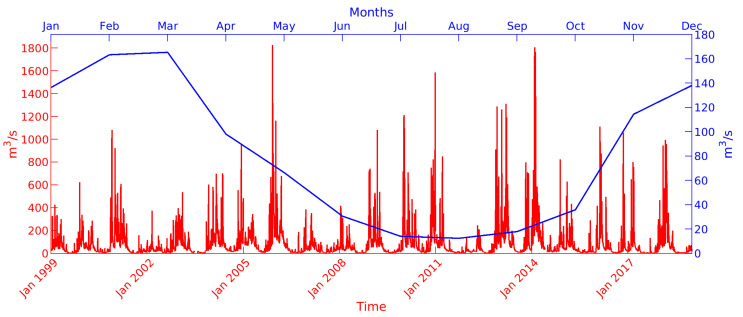
Daily and monthly Arno River discharge values recorded during the period 1999–2018 by the TOS01005191 automatic gauge station managed by the Regional Hydrological Service of Tuscany Region and located close to S. Giovanni alla Vena, approximately 25 km upstream from the river mouth.

**Figure 4 sensors-23-00935-f004:**
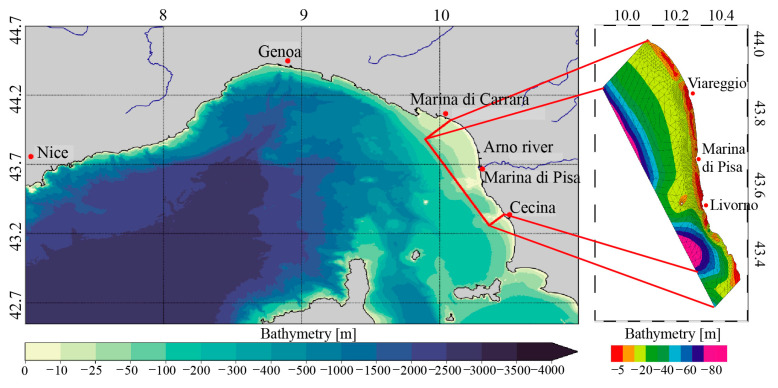
Left panel: study area, with superimposed EMODnet bathymetry; red polygon delimits the boundaries of the SHYFEM model implementation area. Right panel: SHYFEM model domain (horizontal mesh superimposed to bathymetry).

**Figure 5 sensors-23-00935-f005:**
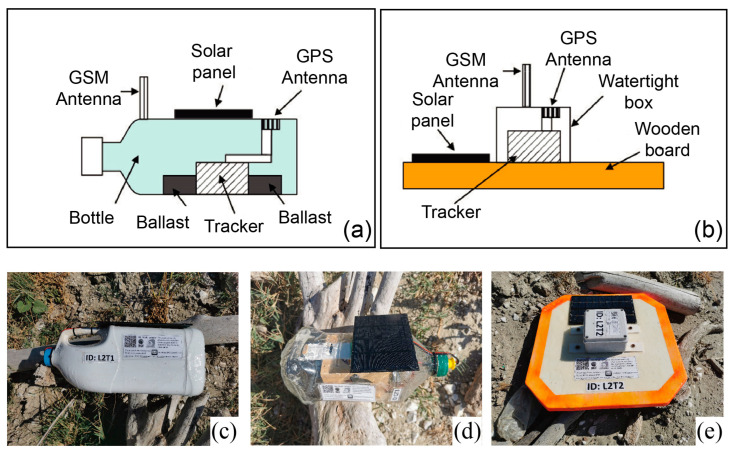
In top panels, the scheme of the two types of MLT (table-shaped type in panel (**a**) and bottle-shaped type in panel (**b**)) and in bottom panels pictures of some of the realized MLTs (two different bottle-shaped types, of different shape and volume, in panels (**c**,**d**) and the table-shaped type in panel (**e**)).

**Figure 6 sensors-23-00935-f006:**
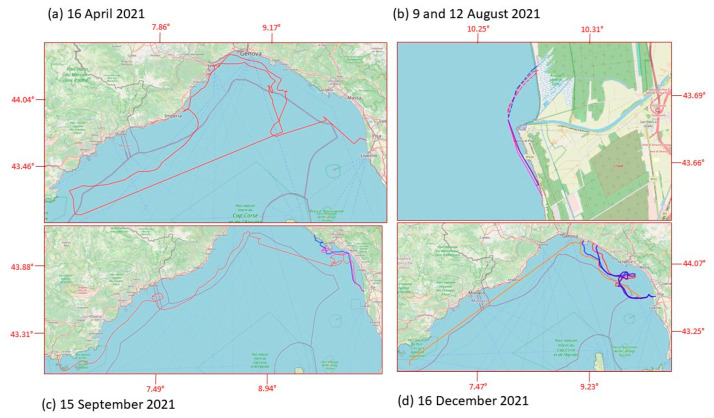
Trajectories of MLT during the 5 launches performed in 2021 (time is expressed according to CET/CEST). Box (**a**) display the launch of 16 April; box (**b**) contains the two launches of 9 (dotted lines) and 12 August; box (**c**) display the launch of 15 September and box (**d**) the one of 16 December. The comparison of the 4 pictures outlines the differences in the dispersion and beaching scenarios of the same type of MLTs in the same coastal area but in different seasons. It is noteworthy that the only MLTs that reached French coastal waters were tablet-shaped.

**Figure 7 sensors-23-00935-f007:**
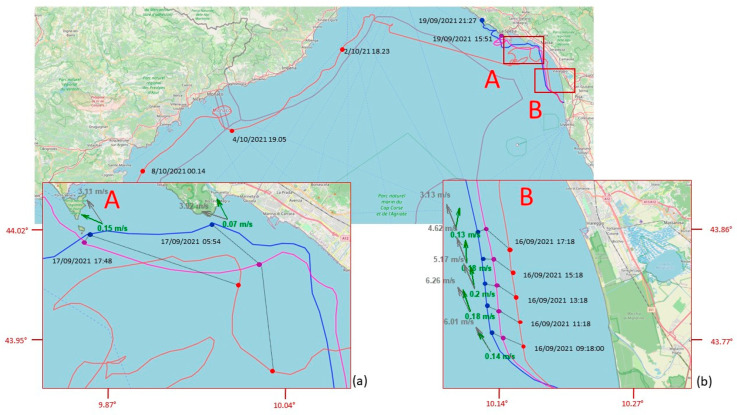
September 2021 launch with three MLT (time is expressed according to CEST). The drifters’ trajectories refer to L3T1 (blue line, bottle-shaped), L3T2 (violet line, bottle-shaped), L2T2 (red line, tablet-shaped), as reported in Table 1. Boxes (**a**) and (**b**) are zooms on the two coastal areas A and B highlighted in the full picture. The green and gray arrows in panels A and B represent current and wind direction, respectively. The associated intensities are reported aside with the same colors. The green arrows indicate wind intensity and direction, the gray arrows for surface current intensity and direction.

**Figure 8 sensors-23-00935-f008:**
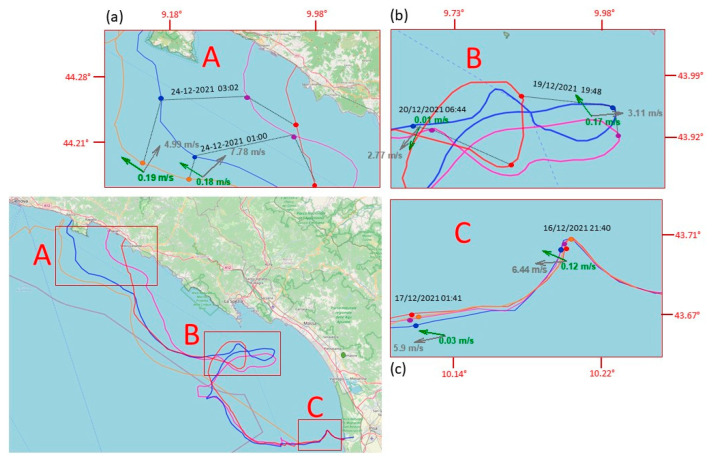
December 2021 launch with four MLTs (time is expressed according to CET). Trajectories correspond to L2T1 (blue line, bottle-shaped); L3T1 (violet line, bottle shaped); L4T3 (red line, table-shaped), L4T1 (orange line, table-shaped), as reported in Table 1. Boxes (**a**), (**b**) and (**c**) are zooms on the three coastal areas A, B and C highlighted in the full picture. The green and gray arrows in these panels represent current and wind direction, respectively. The associated intensities are reported aside with the same colors.

**Figure 9 sensors-23-00935-f009:**
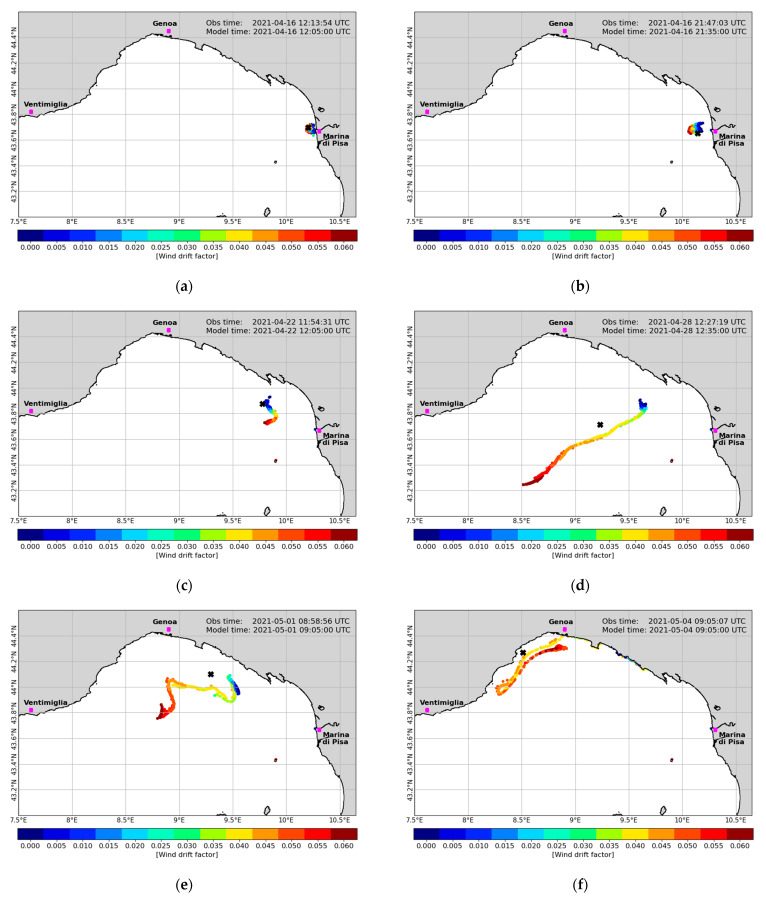
Modeled particles (colored dots), with colors representing different values of wind drift factor (WDF) parameter, compared to the observed position of the tablet-shaped drifter (black cross). Approximate elapsed time after the drifter’s deployment are: 3 h panel (**a**), 12 hours panel (**b**), 6 days panel (**c**), 12 days panel (**d**), 15 days panel (**e**), 19 days panel (**f**), 22 days panel (**g**) and 24 days panel (**h**).

**Table 1 sensors-23-00935-t001:** All launches but the first were performed with 3 to 4 drifters at a time. The prototypes had different shapes (tablet, plastic bottles with 2, 3, and 5 liters volume), and with tracking cards of both the commercial-type and the customized-type. Launch coordinates: latitude 43.68202°, longitude 10.26484° (with uncertainty of 0.005° on both the coordinates). Hour of release: 11:00 CET/CEST (uncertainty of 25 min) for all launches except for the one in September, which started at 19:02 CEST.

Release Date	MLT Id	MLT Type	Tracking Type	Total Track Length (km)	Mean Day Path (Km)	Mean Velocity (Km/h)	Type of Ending-Trip	Number of Days Tracked
16-Apr-21	L1T1	Tablet-Shaped	Commercial WINNIES	852.59	19.91	1.09 +/− 0.74	Beached and lost (UNK ^1^)	42.83
9-Aug-21	L2T1	Bottle-Shaped 3 L	Commercial WINNIES	2.17	n.a.	0.68 +/− 0.50	Beached and recovered (FBP ^1^)	0.14
	L2T2	Tablet-Shaped	Commercial WINNIES	2.81	n.a.	0.81 +/− 0.52	Beached and recovered (FBP ^1^)	0.18
	L2T3	Bottle-Shaped 1,5 L	Customized MADUINO	2.90	n.a.	0.72 +/− 0.66	Beached and recovered (FBP ^1^)	0.17
12-Aug-21	L2T1	Bottle-Shaped 3 L	Commercial WINNIES	3.97	n.a.	0.94 +/− 0.60	Beached and recovered (FBP ^1^)	0.18
	L2T2	Tablet-Shaped	Commercial WINNIES	3.50	n.a.	0.86 +/− 0.60	Beached and recovered (FBP ^1^)	0.17
	L2T3	Bottle-Shaped 1,5 L	Customized MADUINO	3.54	n.a.	1.58 +/− 0.60	Beached and recovered (FBP ^1^)	0.09
15-Sep-21	L3T1	Bottle-Shaped 3 L	Commercial WINNIES	100.18	24.18	1.09 +/− 0.65	Beached and recovered (FBP ^1^)	4.14
	L3T2	Bottle-Shaped 5 L	Commercial WINNIES	102.18	24.17	1.02 +/− 0.68	Beached and recovered (FBP ^1^)	4.23
	L2T2	Tablet-Shaped	Commercial WINNIES	557.58	16.36	0.93 +/− 0.95	Beached and lost (UNK ^1^)	34.09
16-Dec-21	L2T1	Bottle-Shaped 3 L	Commercial WINNIES	185.29	23.29	1.0 +/− 0.39	Beached and lost (UNK ^1^)	7,95
	L3T1	Bottle-Shaped 3 L	Commercial WINNIES	191.89	23.48	1.3 +/− 0.44	Beached and recovered (FBP ^1^)	8.17
	L4T1	Tablet-Shaped	Customized MADUINO	426.93	24.23	1.13 +/− 0.52	Contact lost in open sea (NCC ^1^)	17.62
	L4T3	Tablet-Shaped	Customized MADUINO	160.59	19.81	0.91 +/− 0.68	Beached and recovered (FBP ^1^)	8.11

^1^ FBP = found by public, NCC = no cellular connectivity, UNK = Unknown.

**Table 2 sensors-23-00935-t002:** RMSE and skill score of the Lagrangian model for the deployments of April and December 2021.

	Wind Drift Factor													
		0.00	0.50	1.00	1.50	2.00	2.50	3.00	3.50	4.00	4.50	5.00	5.50	6.00
Deployment Date	Drifter Typology (ID)													
** *16 Apr 2021* **														
*RMSE (km)*	*Tablet shaped (L1T1)*	37.36	35.45	29.60	28.04	26.43	26.13	29.16	35.14	**23.28**	34.58	34.94	44.00	48.43
*SS*	*Tablet shaped (L1T1)*	0.78	0.78	0.79	0.79	0.80	0.80	0.78	0.78	**0.88**	0.84	0.83	0.75	0.71
** *16 Dec 2021* **														
*RMSE (km)*	*Bottle Shaped 3L (L2T1)*	17.47	16.11	14.09	12.14	10.22	8.72	7.59	**6.97**	7.06	7.65	8.16	8.84	9.31
*SS*	*Bottle Shaped 3L (L2T1)*	0.55	0.58	0.62	0.65	0.69	0.72	0.74	**0.76**	0.75	0.74	0.74	0.72	0.71

*RMSE (km)*	*Bottle Shaped 3L (L3T1)*	32.8	30.21	23.96	17.31	**16.22**	24.15	27.39	29.36	30.94	30.94	31.98	32.26	34.22
*SS*	*Bottle Shaped 3L (L3T1)*	0.68	0.71	0.75	0.80	**0.82**	0.79	0.77	0.76	0.75	0.75	0.74	0.74	0.72

*RMSE (km)*	*Tablet shaped (L4T3)*	61.41	51.74	38.13	**24.09**	36.29	41.29	37.51	34.67	34.21	36.86	37.88	38.49	37.80
*SS*	*Tablet shaped (L4T3)*	0.68	0.71	0.81	**0.86**	0.82	0.78	0.79	0.80	0.80	0.78	0.78	0.78	0.76

*RMSE (km)*	*Tablet shaped (L4T1)*	28.56	27.81	27.21	26.07	24.11	19.96	**14.28**	14.49	18.16	19.20	19.54	19.97	20.42
*SS*	*Tablet shaped (L4T1)*	0.56	0.58	0.60	0.61	0.64	0.68	0.74	**0.75**	0.74	0.74	0.74	0.73	0.73

## Data Availability

This study has been conducted using E.U. Copernicus Marine Service data to force the lagrangian model out of the coastal area covered by the high resolution circulation model. (https://doi.org/10.25423/CMCC/MEDSEA_ANALYSISFORECAST_PHY_006_013_EAS6) (accessed on 11 January 2023). The output of the lagrangian model and the tracking positions of drifters are available at Appendix A.

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
