# Peer review of "Marine Litter Tracking System: A Case Study with Open-Source Technology and a Citizen Science-Based Approach"

_sensors, 2023, doi:10.3390/s23020935_

Round 1

Reviewer 1 Report

Manuscript sensors-2095326 is an interesting study, correctly written, with proper structure and includes clear descriptions and explanations of the experimental design. The manuscript contains valuable data and is of great interest. It presents the data in an easy-to-follow manner. 

I  have no negative comments on this work

Author Response

Many thanks for the positive comments. We carried out a review of the English language on the entire manuscript, and particularly in the parts also indicated also by the other reviewers.

We sincerely hope that now the text is more fluent and understandable.

The authors

Reviewer 2 Report

An excellent paper with exhaustive analyses on trash disposal and models for tracking. Authors may consider the following suggestions to add either at the end of Discussion or in Conclusion:

1. methods of involving civil society (citizens, students etc) for monitoring of trash disposal and creating awareness against it,

2. inputs for local authorities for using this study while developing  policies on trash management.

Author Response

Thanks for the comments and the suggestions. We have added and/or clarified the two issues indicated, also including new relevant references. New added sentences can be found in the part of the text between line 1061 and line 1125 of the new version of the manuscript (uploaded).

Moreover, we revised the English language throughout the entire manuscript, and particularly in the parts also indicated by the other reviewers.

We sincerely hope that now the text is more fluent and understandable.

The authors

Reviewer 3 Report

Here are a few suggestions to help with the readability of the paper:

1) At this stage, the title is long and wordy. Can the title be made more concise? E.g., Marine Litter-tracking system: open source technology and citizen science

2) English grammar and expression can be improved. For example, in the abstract, the authors claim that ... 'makers or common citizens can build them'. What is meant by makers or makers' community? Makers of what? Is this a commonly understood phrase in the discipline? Who are 'common citizens'? It is important that the whole article is proof-read before submission

3) Has the citizen science approach previously been used for trash tracking? How is this study's approach different or does it fill a gap?

Author Response

Thanks for the comments and the suggestions. We have changed the title following your suggestion. We have specified the meaning of “makers” in a dedicated note (Note 2 on page 29). Moreover, we added and/or clarified the topics included in the two questions contained in the point 3 of your comments. In particular, the answers to these two questions are reported in the paragraph from line 1061 and line 1125 of the "Discussion", in the new manuscript version. We also included new relevant related references.

Moreover, we revised the English language throughout the entire manuscript, and particularly in the parts also indicated by the other reviewers.

We sincerely hope that now the text is more easily fluent and understandable.

The authors